# A Study on Project Prioritisation and Operations Performance Measurements by the Analysis of Local Financial Investment Projects in Korea

**Heecheol Shim and Jaehwan Kim \***

Department of Real Estate Studies, Kongju National University,
Yesan-gun 32439, Chungcheongnam-do, Republic of Korea
\* Correspondence: jaehwan@kongju.ac.kr; Tel.: +82-41-330-1402

**Abstract:** The study aims to prioritise investment projects of local Korean governments through priority analysis and provide basic data to secure sustainability by measuring the operational performances of these projects. We utilised the AHP technique to draw relative weight, followed by their correction, thereby deriving priorities across all 14 fields; the most important ones being social welfare (0.110), healthcare (0.108), fire prevention (0.097), environmental hygiene (0.092), and cultural tourism (0.088). Important areas prioritise balance and equity and their operational efficiency and productivity were based on statistical annual reports of 17 cities and provinces in Korea and Kosis data obtained from the National Statistical Office. For the measurement, we used six years of time series data from 2015 to 2020 and single point-in-time and annual trend analyses. The input factors were the amount of revenue and the number of public officials, and the output factors included the water and sewage supply rate, the number of tourists travelling for free, and the number of hospitals, welfare facilities, and rescue cases. These selections were appropriate to draw conclusions by analysing operational performance with detailed items in the priority field. Results revealed that the operating performance was excellent. However, to secure additional operational efficiency and productivity, technological innovation is needed.

**Keywords:** local finance; investment business; analytical stratification process; operational performance measurement; productivity; efficiency

## 1. Introduction

### 1.1. Background and Purpose of Study

Most investment projects by local governments requiring large-scale funding and resources significantly impact local finances and the lives of local residents [1]. In recent years, various projects have been carried out considering regional characteristics to achieve balanced regional development with more emphasis on social infrastructure as opposed to economic infrastructure [2]. Therefore, the Korean government operates an examination system for local governments' investment projects in accordance with Articles 36 and 37 of the Local Finance Act, Article 41 of the Enforcement Decree of the Local Finance Act, and the Local Finance Investment Project Review Rules [3]. The investment review system plays a role in suppressing budget overlap and waste by analysing and evaluating project feasibility in advance and selecting appropriate investment projects. Therefore, an investment review is usually conducted before budgeting and is divided into self-examination and requested review according to the funding scale and local ordinance. Investment review shall be conducted by holding an Investment Review Committee, and in the case of appropriate projects, funding support is made starting with budgeting to streamline the implementation of the project. For the projects rejected or in review, support is rendered to secure economic feasibility [4]. Investment screening is gaining importance in Korea due to a decrease in revenue caused by a fall in population and the need for balanced regional development,

and also for the prevention of indiscriminate project promotion [5]. Therefore, investment reviews based on clear, objective standards help maximise the effective use of resources [6].

However, the current investment review applies only one criterion to all business areas, so the specificity is insufficient, and the issues of inequality and inconsistency are raised due to the subjective and arbitrary evaluation and failure to apply weight according to regional characteristics [7]. In recent years, different sectors appeared to have different characteristics due to various project types and their complex aspects, and concerns prevail that side effects may ensue without clear prioritisation [8]. The biggest limitation is poor follow-up management. Even if the investment review is passed, it is rare to proceed with the original plan, and continuous local expenses are invested in the future operation process, so it should be properly managed and controlled [9]. However, there is no system that can clearly manage or monitor how the project is going after the investment review. Therefore, it is necessary to monitor the progress of the project in the pre-budget and the subsequent stage [10]. In fact, 21 investment projects led by Korean local governments return an average of more than KRW 5 billion in revenue annually [11]. Since they are public development projects, they cannot be judged based on simple profitability; however, if properly managed, effective resource distribution can be achieved.

Therefore, in this study, an analysis model is constructed as follows and implications are presented (Table 1). First, we intend to derive priorities by measuring the importance of each field of local investment projects by evaluators and relevant officials. Second, based on the derived priorities, this study will determine how resources are actually allocated and operated with efficiency and productivity. Third, based on the results of the analysis, this study will examine the performance of the prioritised areas in the operation stage, suggesting areas to improve or implications.

**Table 1.** Analysis model.

| | |
|---|---|
| Derivation of Priority Businesses | - Conduct a survey with a group of experts based on the finalised hierarchical structure<br>- The analysis uses AHP analysis to measure the relative importance<br>- Identify priority business areas based on derived priorities |
| Efficiency and Productivity Analysis of 17 Cities | - Measurement of efficiency and productivity of 17 cities and provinces in Korea for priority areas<br>- Measurement is performed based on data from 2015 to 2020<br>- Extend the limits of static analysis to further perform trend analysis |
| Results and Conclusions | - Through results of efficiency and productivity analysis of projects, measures of improving future investment reviews are presented, along with policy and academic implications |

### 1.2. Scope and Method of Research

This study derived the fields of each project through industry-related tables and existing investment review history management, and reclassified them based on previous studies. Based on the classified fields, a major classification of economic and social infrastructure was added to suit the current trend of investment projects in Korea, and the items of equity, effectiveness, balance, and efficiency were classified into first and second priority to examine both public interest and profitability for operational sustainability. Based on a total of five groups classified in this method, the field of investment review was classified into fourteen fields. This is a hierarchical structure diagram reviewed by experts using FGI. Based on this, the relative importance was measured using the hierarchical analysis to correct and determine each priority level. The main areas presented as priorities were put to efficiency and productivity analysis based on the statistical yearbook of 17 cities and provinces in Korea and the Kosis data of the National Statistical Office. Through this, we determined the extent to which the fields selected as major projects in the investment review were effectively utilising resources in the actual operation stage. The efficiency of DMU for each unit was measured, and the Malmquist index was derived to determine

the size of input elements and to present a plan for production scale. Therefore, this study aims to measure operational performance with the efficiency and productivity of the areas derived as priorities and to present proposals and implications for the most effective use of resources.

## 2. Theoretical Considerations

### 2.1. Overview of Local Financial Investment Projects

Local financial investment projects are reviewed before budgeting, and projects that are invested in kind are reviewed for the necessity and feasibility of the project plan before implementing the project. The screening procedures are shown in Table 2.

**Table 2.** Local financial investment project review procedure.

| Mid-term Regional Fiscal Planning | → | Feasibility Study by the Local Finance Act | → | Local Finance Investment Review | → | Application for Subsidiaries (Issuance of Local Bonds if Necessary) | → | Budgeting of Local Governments | → | Feasibility Reviews of other Laws such as the Development Act | → | Promotion and Execution of Projects | → | Analysis of Local Finances |
|---|---|---|---|---|---|---|---|---|---|---|---|---|---|---|

Local fiscal investment review is largely divided into projects with budgets for every detailed item and event projects in terms of expenditure budget structure. The subject of the review is the new investment projects planned to be implemented in the next fiscal year. An investment review shall be commissioned within this period from the establishment of the basic plan until the implementation of the implementation design. Investment screening is conducted thrice a year, and in special cases, an occasional screening is carried out. The schedules for regular screening are shown in Table 3.

**Table 3.** Local financial investment project review schedule.

| Reviews | Date of Request | Date of Review |
|---|---|---|
| First Review | Until 1.1. (12.15., prior year) | End of 02.28. |
| Second Review | Until 3.31. (3.15.) | End of 5.31. |
| Third Review | Until 6.15. (5.31.) | End of 8.15. |
| Fourth Review | Until 8.25. (8.10.) | End of 10.25. |

Note: Screening can be extended within 40 days
Data: Manual for the Review and Feasibility Study of Local Financial Investment Projects (09,13,22)

In the case of cities and provinces, the first screening schedule is required to be submitted to the Minister of Public Administration and Security by January 31, the second by April 30, and the third by August 31. In the case of cities, counties, and districts, the first screening schedule is required to be submitted to the city and provincial governors by January 15, the second by April 15, and the third by August 15. The occasional review schedule can be submitted to the Minister of Public Administration and Security through the city and provincial governors. The results of the investment review are to be reported within 14 days from the date of its completion. The review results are used to determine financial support and budget [12].

### 2.2. AHP Theory

The analytic hierarchy process (AHP), which is used as one of the multi-standard decision-making methods, was developed by T. Saaty (1980). It seeks to capture intuition based on evaluators' expertise, experience, and judgments on various factors through pairwise comparisons between elements of a set hierarchy. It performs tasks to solve problems by comparing factors against each other for common purposes or criteria with the analytical process of judgment and an analytical process through the establishment of a hierarchy being essential factors. AHP has recently been widely used for decision making,

having the advantages of theory simplicity, convenience of application, and versatility by utilising the intuitive and reasonable judgment of decision makers and the empirical data they have accumulated. It can measure values by considering qualitative and quantitative aspects together, and it can facilitate evaluation through pairwise comparison between hierarchical elements. Even in pairwise comparisons, it has the advantage of easy applicability because it is relatively free to grant the scale. It can also verify logical consistency to ensure the reliability of results, increase objectivity by hierarchically representing listed complex decision-making problems, and analyse sensitivity to predict decision-related issues and changes in information and circumstances. It effectively supports the analysis of decisions made collectively by multiple people.

It offers the following advantages compared to the existing decision-making method. First, it is possible to measure through a ratio scale by considering both qualitative and quantitative criteria in the decision-making process. Through this, it is possible to support decision making more rationally and realistically as qualitative factors are considered together with quantified data in the decision-making process. Second, by differentiating evaluation factors based on a hierarchical structure diagram, a measure can be prepared to solve objective problems of complex social and political matters. The hierarchical structure diagram can provide the basis for complex decision-making processes for practical areas based on previous studies and theories, and simulations can be conducted on the final choice alternatives assigned to each element. Third, it is easy to collect various opinions from experts. AHP provides an integrated criterion for collective decision making, thereby preventing domination and distortion that can occur in the decision-making process, enabling more objective decision-making [13].

The decision-making process of the AHP technique is as follows. First, it hierarchically expresses unstructured and sporadically distributed decision-making problems. Thus, all the elements that make up the decision-making field are listed and classified into different levels. Similar elements at the same level are collected together to be sorted, followed by structuring and systematising things around the problems. Through this, it is possible to accurately identify problems related to decision making. Second, the matrix of the pairwise comparison for each level is obtained through the framework of the pairwise comparison. Weight granted through pairwise comparison refers to relative importance or weight. In the evaluation process by pairwise comparison, decision makers' preferences of evaluation criteria are first expressed by semantic expression, and the quantification process of granting numerical values for correction is accompanied. To this end, the 9-point scale proposed by Saaty is commonly used [14].

### 2.3. DEA Model and Malmquist Productivity Index Model

The data envelopment analysis (DEA) technique is a method of measuring relative efficiency in the management and operation of non-profit decision-making units (DMUs) that perform the same or similar functions using multiple inputs and outputs based on a system model. This model is the most appropriate way to evaluate inefficiency in the public sector where market prices are not formed [15]. The model, in which one of the inputs and outputs is fixed, is divided into an input-oriented model and an output-oriented model depending on how to find inefficient areas in the remaining elements. Additionally, depending on whether the effect of scale is considered in measuring efficiency, it can be divided into the CCR and BCC models. The former assumes constant returns to scale (CRS) of the production available set, and the latter assumes variable returns to scale (VRS). The Malmquist productivity index is calculated using a distance function. If the DEA model builds a production frontier and obtains the efficiency of individual public institutions or companies, the Malmquist productivity index can be derived as an input or output orientation of the distance function itself [16]. This study sets an output-oriented distance function that maximises input with output constant. Through this, we intend to use a formula that summarises the production technology of stage $t$ and the production technology of stage ($t + 1$) through estimation analysis for the change in productivity.

*2.4. Review of Previous Studies*

Until now, research that directly conducted the priorities of existing public development projects and their efficiency and productivity have been scarce [17]. This is because each project field has a different purpose and direction to pursue. In the case of public development projects conducted in Korea, location selection tends to be determined by political standards or unilateral judgments by administrative offices. In the case of efficiency and productivity analysis, studies conducting measurements annually to report exist; however, studies that measured changes over time through trend analysis are insufficient.

Therefore, this study intends to review previous studies by classifying investment screening fields into studies classified by type and those analysing the operation status of local governments in terms of efficiency and productivity. First, the studies that classified and standardised investment screening according to the type of target projects are as follows. Local financial investment projects are regional development projects that include local government funds due to their nature, and relevant research conducted considers efficiency, equity, and environmental conservation in addition to the classification of projects [18]. Thus, efficiency is also considered for public development projects. Additionally, there is a study that categorises investment target projects by function. First, studies classified by social functions are classified into daily projects, auxiliary projects, developmental projects, and basic social projects, which means that social functions are evaluated with some variation between fields. From a similar point of view, public investment projects are sometimes classified into economic infrastructure such as roads, power, water and sewage, bridges, and social infrastructure such as childcare, environment, and hygiene. These projects can be divided into economic and social facilities in terms of macroeconomic development because economic facilities are considered important in terms of efficiency and social infrastructure in terms of balance [19]. Furthermore, target projects are classified into production and purchase entities. Thus, projects are classified based on whether the producer stood out significantly in the public or private interest, and the types of service purchased [20]. After all, this is very similar to classifying the project into basic service projects including roads and sewage treatment plants and lifestyle improvement projects involving libraries [21]. This is similar to Hansen's project classification [20]. The study classifies the criteria for investment review based on the scope of supply costs and benefits, focusing on the influence of a project, in line with the current investment review, in which investment reviewers change based on the size of the total project funding [22]. I.M. Barlow [23] also categorises public investment according to the magnitude of the ripple effect like W. F Smith [22] does. Another study by him pointed out effectiveness, efficiency, responsiveness, equity, and appropriateness as major determinant criteria for the evaluation and analysis of public investment projects. Additionally, he also argued that equity and effectiveness should be considered in the case of urban infrastructure classified as basic infrastructure facilities.

Second, studies that analysed the operation status of local governments through efficiency and productivity analysis can be divided into a macro approach and a micro approach [24]. This is because efficiency or productivity analysis has a very large influence on the analysis of the type or characteristics of the input variable. Therefore, there is a need to secure objectivity in variable selection and clarify categories. As a result of reviewing previous studies based on this, this study found a study that identified the efficiency index by dividing it into distributed efficiency, technical efficiency, and cost efficiency using DEA for local governments in Australia. As a result of Tobit regression analysis using independent variables such as debt repayment cost, subsidy dependence, number of public officials per 1000 population, and current asset amount, the subsidy dependence did not significantly affect any efficiency [25]. Another study analyses the change in efficiency by era and single-term efficiency of autonomous districts by applying DEA and DEA/window analysis based on data for five years from 2003 to 2007 for 15 autonomous districts in Busan in Korea. It proposes a model that can analyse the extent to which the efficiency of autonomous districts is changing by era. The measurements revealed that the efficiency of

autonomous districts, from Windows 1 to 3, showed a continuous decline with time except for a few districts [26]. The study that attempted the second stage Boostrap-DEA used data from 75 city governments in 2006 and 2008. Results revealed that the total amount of local grant tax and efficiency share a negative relationship. However, contrary to the opinion of independent finance, there was no significant relationship between the total amount of state subsidies and efficiency [27]. A study that evaluated financial efficiency for integrated cities was conducted on 37 local governments out of 41 cities integrated in 1995. The goal of increasing fiscal efficiency by expanding the population of local governments through urban and rural integration and thus lowering the marginal cost of supply services was slightly achieved in the short term. However, the rate of increase in population size at the time of integration was slower than the rate of increase in the minimum efficient population size, and in the long run, the increase in fiscal efficiency through economic realisation at the time of integration was not achieved [28]. In the end, to improve the fiscal efficiency of local governments, these studies sought to find ways to streamline fiscal expenditure by categorising inefficiency in fiscal activities and presenting implications for realising economies of scale. As a result of measuring using the BCC-VRS model and the CCR-CRS model, we also found studies showing that cities with large population sizes and higher financial independence have higher fiscal efficiency. To increase fiscal efficiency, it is necessary to have a system that can systematically manage all processes from budget planning to striking contracts, execution, and follow-up management to improve fiscal efficiency by strengthening cooperation among local governments. Budget cuts and economies of scale are required [29].

Most of the preceding studies were conducted simply to distinguish fields or focused only on efficiency and productivity at a single point in time. Moreover, no studies have undergone a process of converging, analysing, and verifying the two fields together. Therefore, this study is judged to be able to present improvement measures to secure the soundness of local finances and prevent duplicate investments through the conclusions drawn. It is believed that it will be able to support the successful promotion of projects through various derivative effects such as deriving necessary projects in regions, prioritising projects, and selecting places for the projects. Additionally, this analysis result can be used as basic data to diversify operation methods and the feasibility of development projects regarding operation type.

## 3. Results of Deriving Priorities for Local Financial Investment Projects

To derive priorities, the investment project field was divided into a total of 14 fields, and the AHP technique was used to analyse them. We reviewed various data on the existing classification table for each project field and previous domestic and foreign studies. Finally, the final hierarchical structure diagram was confirmed based on the in-depth FGI interview by the expert group. In the case of the first tier, the type of public investment projects was selected to cover projects such as roads, electricity, water supply, bridges, and social infrastructure such as education, health hygiene, fire fighting, police, and welfare for the elderly. This is consistent with Thurmond (1989) [30], who argued that once a certain level of demand for economic infrastructure is met, the share of investment in social infrastructure gradually shifts to the next. In the case of the second tier, it is a question of how to set the target value, and when categorised, only one single intrinsic value cannot be seen, so the result of mixing it should be presented. Dunn (1981) [31] suggested effectiveness, efficiency, equity, and appropriateness. In addition, various scholars have additionally suggested effectiveness, productivity, and appropriateness. Among them, the final mixed value was selected based on the FGI results in accordance with the situation in Korea, and numbers indicate priority for importance. In the case of the third tier, the final structural diagram was established by organising the parts corresponding to public development projects among the items of the large classification based on the industry-related table published in the Republic of Korea. The final derived hierarchical structure diagram is as follows (Table 4).

**Table 4.** Finalised hierarchical structure.

| Evaluation of Local Financial Investment Projects | | | | | | | | | | | | | |
| --- | --- | --- | --- | --- | --- | --- | --- | --- | --- | --- | --- | --- | --- |
| **Economics Infrastructure** | | | | | | | **Social Infrastructure** | | | | | | |
| 1. Equity<br>2. Effectiveness | | | 1. Balance<br>2. Efficiency | | | | 1. Balance<br>2. Equity | | 1. Efficiency<br>2. Equity | | | 1. Efficiency<br>2. Balance | |
| Basic Facility | Water Supply and Sewage | Environmental Hygiene | Road Traffic | Industry/SMEs | Regional Development | Agriculture, Forestry and Fisheries | Healthcare | Social Welfare | Parks | Fire Prevention | Culture and Tourism | Education and Sports | General Administration |

Note: This hierarchical structure map revised and supplemented Shim Hee-cheol's (2020) [17] study on the application of weights by field in the review of local financial investment projects.

The sample comprised 107 copies, and 102 copies were used for the analysis, excluding 5 samples with a consistency index of 0.1 or higher. A total of 42 university professors (41.2%), 33 from government-funded research institutes (32.4%), 15 from private architectural firms or engineering companies (14.7%), and 12 other expert groups or public officials (11.7%), participated in the investment review. All of them are investment project judges, managers, or researchers specialising in feasibility studies, and they have more expertise in this field of study than anyone else. The final analysis results are shown in Table 5.

**Table 5.** Priority derivation results.

| Class 1 | AHP | Class 2 | AHP | Class 3 | AHP | Correction Value | Priority |
|---|---|---|---|---|---|---|---|
| Economic Infrastructure | 0.398 | 1. Equity 2. Effectiveness | 0.571 | Basic Facilities | 0.295 | 0.067 | 9 |
| | | | | Water Supply and Sewage | 0.301 | 0.068 | 7 |
| | | | | Environment and Hygiene | 0.404 | 0.092 | 4 |
| | | 1. Balance 2. Efficiency | 0.429 | Road and Traffic | 0.252 | 0.043 | 12 |
| | | | | Industry/SMEs | 0.294 | 0.050 | 10 |
| | | | | Regional Development | 0.238 | 0.041 | 13 |
| | | | | Agriculture, Forestry and Fisheries | 0.216 | 0.037 | 14 |
| Social Infrastructure | 0.602 | 1. Balance 2. Equity | 0.361 | Healthcare | 0.495 | 0.108 | 2 |
| | | | | Social Welfare | 0.505 | 0.110 | 1 |
| | | 1. Efficiency 2. Equity | 0.274 | Parks | 0.411 | 0.068 | 7 |
| | | | | Fire Prevention | 0.589 | 0.097 | 3 |
| | | 1. Efficiency 2. Balance | 0.365 | Culture and Tourism | 0.402 | 0.088 | 5 |
| | | | | Education and Sports | 0.383 | 0.084 | 6 |
| | | | | General Administration | 0.215 | 0.047 | 11 |

The analysis revealed that the recent trend of investment projects in Korea was reflected well in the priority derived. When the priority, according to the final correction value, was divided into five areas, social welfare (0.110), healthcare (0.108), fire prevention (0.097), environmental hygiene (0.092), and culture and tourism (0.088) showed the highest priorities. These results showed the difference between the time periods before and after COVID-19. Social welfare and healthcare for vulnerable groups and medical facilities were ranked quite high, followed by safety-related fire prevention and environmental hygiene. This result is believed to be sufficiently available as basic data for future revenue reduction, balanced regional development, and priority project derivation.

## 4. Efficiency and Productivity Analysis Results

The previous analysis involved DEA analysis and the Malmquist index measurement to confirm the degree of efficiency and productivity in actual operation for the five priority areas. The analysis period was set to six years from 2015 to 2020 to identify trend changes. Looking at previous studies, there is a study [32] that estimates the input factors as the number of public officials, labour costs, capital, and revenue, and the output factors as water supply, local tax collection, and the number of residents. In addition, there are

studies [33,34] that select the number of public officials per citizen as input factors and estimate the building permit area, sewage supply rate, water supply rate, and road rate as output factors. Finally, the final variables were selected based on various previous studies, such as the number of general public officials, political officials, technical officials, and local taxes, including sewage penetration rate, water supply rate, number of public sports facilities, and number of welfare facilities. As a result, the input factors included the amount of revenue (KRW million) and the number of public officials (persons) based on the results of reviewing previous studies. The reason for selecting the amount of revenue, and not the amount of expenditure, is that most local fiscal investment projects are large-scale projects, and the issuance of local bonds accounts for a large portion. Moreover, the number of public officials (persons) calculated as input factors in other studies was added and finalised. The final calculation variables include the water supply rate (%), the sewage supply rate (%), the number of tourists (persons), the number of hospitals (number), the number of welfare facilities (number), and the number of rescue cases (number). The most representative variables were selected for calculation by reflecting the previous five priority areas: social welfare, healthcare, fire prevention, environmental hygiene, and cultural tourism. Additional variables may be included, but the reliability of efficiency and productivity analysis significantly drops if input variables are more than twice the total DMU. Thus, the number of variables was set to ensure the reliability and objectivity of analytical results [35]. The basic statistics derived prior to the analysis results are shown in Table 6.

**Table 6.** Basic statistics.

| Year | Category | Minimum Value | Maximum Value | Average | Standard Deviation |
|------|----------|---------------|---------------|---------|---------------------|
| 2015 | Tax Revenue (KRW 100 Mil) | 1,301,680.00 | 41,638,559.00 | 15,701,614.65 | 10,121,247.15 |
| | Number of Officials | 1437.00 | 38,214.00 | 13,851.41 | 9624.48 |
| | Water Supply Rate (%) | 86.60 | 100.00 | 95.30 | 4.90 |
| | Sewage Supply Rate (%) | 74.35 | 100.00 | 90.68 | 8.49 |
| | Number of Tourists Travelling for Free | 463,085.00 | 54,731,858.00 | 8,964,062.41 | 12,824,629.26 |
| | Hospitals | 202.00 | 13,338.00 | 3035.00 | 2933.83 |
| | Welfare Facilities | 428.00 | 9469.00 | 3900.41 | 3155.80 |
| | Rescue Cases | 2065.00 | 109,767.00 | 25,302.88 | 25,402.27 |
| 2016 | Tax Revenue (KRW 100 Mil) | 1,799,178.00 | 47,131,165.00 | 17,050,910.82 | 11,125,307.35 |
| | Number of Officials | 1568.00 | 39,064.00 | 14,265.71 | 9813.69 |
| | Water Supply Rate (%) | 87.70 | 100.00 | 95.79 | 4.37 |
| | Sewage Supply Rate (%) | 76.92 | 100.00 | 91.06 | 7.91 |
| | Number of Tourists Travelling for Free | 778,591.00 | 54,401,829.00 | 10,144,600.94 | 12,611,528.15 |
| | Hospitals | 245.00 | 13,983.00 | 3119.53 | 3075.71 |
| | Welfare Facilities | 453.00 | 9545.00 | 3928.47 | 3170.89 |
| | Rescue Cases | 4195.00 | 115,724.00 | 31,236.88 | 26,325.95 |

**Table 6.** *Cont.*

| Year | Category | Minimum Value | Maximum Value | Average | Standard Deviation |
|---|---|---|---|---|---|
| 2017 | Tax Revenue (KRW 100 Mil) | 1,726,718.00 | 51,574,205.00 | 17,785,927.59 | 12,041,570.15 |
| | Number of Officials | 1355.00 | 40,618.00 | 14,367.59 | 10,121.94 |
| | Water Supply Rate (%) | 89.00 | 100.00 | 96.42 | 3.79 |
| | Sewage Supply Rate (%) | 77.90 | 100.00 | 91.97 | 7.77 |
| | Number of Tourists Travelling for Free | 1,221,858.00 | 55,343,260.00 | 11,573,825.24 | 13,178,999.40 |
| | Hospitals | 300.00 | 14,174.00 | 3186.94 | 3113.92 |
| | Welfare Facilities | 462.00 | 9667.00 | 3962.59 | 3201.22 |
| | Rescue Cases | 5383.00 | 143,028.00 | 32,922.41 | 31,820.85 |
| 2018 | Tax Revenue (KRW 100 Mil) | 1,874,648.00 | 56,100,312.00 | 18,930,651.24 | 13,112,407.55 |
| | Number of Officials | 1758.00 | 41,016.00 | 15,016.59 | 10,172.28 |
| | Water Supply Rate (%) | 89.80 | 100.00 | 96.85 | 3.42 |
| | Sewage Supply Rate (%) | 79.70 | 100.00 | 92.61 | 7.00 |
| | Number of Tourists Travelling for Free | 1,354,853.00 | 51,944,448.00 | 13,719,795.29 | 15,852,797.39 |
| | Hospitals | 328.00 | 14,211.00 | 3221.76 | 3115.23 |
| | Welfare Facilities | 457.00 | 9834.00 | 4001.53 | 3233.64 |
| | Rescue Cases | 5881.00 | 141,050.00 | 33,394.71 | 31,006.15 |
| 2019 | Tax Revenue (KRW 100 Mil) | 1,911,976.00 | 62,153,319.00 | 21,374,607.65 | 14,495,897.79 |
| | Number of Officials | 1817.00 | 37,775.00 | 15,260.24 | 9767.58 |
| | Water Supply Rate (%) | 90.70 | 100.00 | 97.17 | 3.20 |
| | Sewage Supply Rate (%) | 80.82 | 100.00 | 92.76 | 6.51 |
| | Number of Tourists Travelling for Free | 1,328,842.00 | 47,885,119.00 | 12,392,853.00 | 12,310,955.42 |
| | Hospitals | 351.00 | 15,111.00 | 3303.59 | 3312.84 |
| | Welfare Facilities | 465.00 | 10,054.00 | 4049.12 | 3273.96 |
| | Rescue Cases | 6465.00 | 128,830.00 | 34,911.00 | 28,813.38 |
| 2020 | Tax Revenue (KRW 100 Mil) | 2,135,782.00 | 67,132,902.00 | 23,284,645.82 | 15,328,420.12 |
| | Number of Officials | 1848.00 | 40,245.00 | 15,347.47 | 10,248.73 |
| | Water Supply Rate (%) | 92.00 | 100.00 | 97.38 | 2.91 |
| | Sewage Supply Rate (%) | 82.00 | 100.00 | 93.59 | 5.66 |
| | Number of Tourists Travelling for Free | 951,885.00 | 24,632,227.00 | 8,397,170.29 | 7,331,913.45 |
| | Hospitals | 370.00 | 15,856.00 | 3378.88 | 3479.58 |
| | Welfare Facilities | 470.00 | 10,082.00 | 4060.06 | 3276.51 |

Technology efficiency is inefficient if it is close to zero and efficient if it is close to one. As a result of the analysis in Table 7, it was found that the overall efficiency decreased from an average of 93% in 2015 to 88% in 2020. DMUs 8, 9, 10, 14, 16, and 17 have been efficiently operated for five years. Most local governments maintained an average efficiency of 0.9 or higher until 2019; however, it plummeted in 2020, likely due to the influence of COVID-19.

**Table 7.** CCR efficiency analysis result.

| DMU | 2015 | 2016 | 2017 | 2018 | 2019 | 2020 |
|---|---|---|---|---|---|---|
| 1 | 1.0000 | 1.0000 | 1.0000 | 1.0000 | 0.8947 | 0.7573 |
| 2 | 0.8041 | 0.8407 | 0.8307 | 0.8229 | 0.7403 | 0.6648 |
| 3 | 0.8534 | 0.9193 | 0.8892 | 0.8957 | 0.8534 | 0.8339 |
| 4 | 0.7808 | 0.8194 | 0.7538 | 0.7012 | 0.6037 | 0.5338 |
| 5 | 0.8940 | 1.0000 | 0.9295 | 0.9261 | 0.8028 | 0.7352 |
| 6 | 0.9500 | 0.9474 | 0.9005 | 0.9409 | 0.8507 | 0.9506 |
| 7 | 0.8638 | 1.0000 | 0.8017 | 0.8343 | 0.8588 | 0.8285 |
| 8 | 1.0000 | 1.0000 | 1.0000 | 1.0000 | 1.0000 | 1.0000 |
| 9 | 1.0000 | 1.0000 | 1.0000 | 1.0000 | 1.0000 | 1.0000 |
| 10 | 1.0000 | 1.0000 | 1.0000 | 1.0000 | 1.0000 | 1.0000 |
| 11 | 0.8954 | 0.9086 | 0.9134 | 0.9432 | 0.9416 | 1.0000 |
| 12 | 1.0000 | 0.8761 | 0.8354 | 0.9087 | 1.0000 | 0.9564 |
| 13 | 1.0000 | 1.0000 | 1.0000 | 1.0000 | 1.0000 | 0.9855 |
| 14 | 1.0000 | 1.0000 | 1.0000 | 1.0000 | 1.0000 | 1.0000 |
| 15 | 0.7956 | 0.8160 | 0.8051 | 0.8618 | 0.8798 | 0.7572 |
| 16 | 1.0000 | 1.0000 | 1.0000 | 1.0000 | 1.0000 | 1.0000 |
| 17 | 1.0000 | 1.0000 | 1.0000 | 1.0000 | 1.0000 | 1.0000 |
| Average | 0.9316 | 0.9487 | 0.9211 | 0.9315 | 0.9074 | 0.8825 |

Table 8 presents a comparison of the model efficiency of output-oriented scale income invariant (CRS) and scale income variable (VRS) data capture analysis. We classified the analysis values into technology efficiency (TE), pure technology efficiency (PTE), and scale efficiency (SE). A value close to 1 for each item means that it is completely efficient, and a value close to 0 indicates that it is an inefficient input.

**Table 8.** BCC efficiency analysis result.

| Category | 2015 | 2016 | 2017 | 2018 | 2019 | 2020 | Average |
|---|---|---|---|---|---|---|---|
| TE | 0.9316 | 0.9487 | 0.9211 | 0.9315 | 0.9074 | 0.8825 | 0.9205 |
| PTE | 0.9837 | 0.9794 | 0.9841 | 0.9824 | 0.9771 | 0.9696 | 0.9794 |
| SE | 0.9475 | 0.9688 | 0.9363 | 0.9481 | 0.9285 | 0.9084 | 0.9396 |
| N | 17 | 17 | 17 | 17 | 17 | 17 | 17 |

Looking at the average values from 2015 to 2020, TE was 0.9205, PTE was 0.9794, and SE was 0.9396. This indicates that the most efficient operation is possible when the level of maintaining the status quo is derived while reducing the input amount to an average of 92.05% assuming CRS, and 97.94% assuming VRS. The results show that the top 90% is derived as an important area in the investment review, indicating that great efforts are being made in the operation stage. In detail, TE and PTE in 2020 were the lowest. This is judged to be the result of a decrease in tourists due to COVID-19 and a lack of medical facilities. As for the scale income, the increasing return, the diminishing return, and the invariant state were analysed, and the analysis results are shown in Table 9.

**Table 9.** Results of scale earnings analysis.

| Category | 2015 | 2016 | 2017 | 2018 | 2019 | 2020 | Total (%) |
|---|---|---|---|---|---|---|---|
| CRS (Constant Return to Scale) | 9 | 10 | 8 | 8 | 8 | 7 | 50 (49%) |
| DRS (Decreasing Return to Scale) | 7 | 6 | 9 | 9 | 9 | 10 | 50 (49%) |
| IRS (Increasing Return to Scale) | 1 | 1 | 0 | 0 | 0 | 0 | 2 (2%) |
| N | 17 | 17 | 17 | 17 | 17 | 17 | 102 (100%) |

Finally, the MPI was measured to determine the degree of change in efficiency and productivity between points of time by conducting a six-year analysis. This is an analysis value presented based on CRS. MPI represents productivity according to the change in the point of time between the two. If the MPI is greater than 1, it can be seen that the productivity increased between the t and t + 1 points of time; if it is 1, there is no change, and if it is lower than 1, it indicates a decrease. The analysis results are shown in Table 10.

**Table 10.** Malmquist productivity index change rate.

| Category | TECI | TCI | PECI | SECI | MPI |
|---|---|---|---|---|---|
| t2 (2015–2016) | 1.0196 | 1.0229 | 0.9956 | 1.0241 | 1.043 |
| t3 (2016–2017) | 0.9693 | 1.0511 | 1.0051 | 0.9644 | 1.0189 |
| t4 (2017–2018) | 1.0112 | 0.9368 | 0.9971 | 1.0142 | 0.9473 |
| t5 (2018–2019) | 0.9702 | 0.973 | 0.9926 | 0.9774 | 0.9440 |
| t6 (2019–2020) | 0.9663 | 0.9302 | 0.9913 | 0.9748 | 0.8988 |
| Geometric Mean | 0.9871 | 0.9817 | 0.9963 | 0.9907 | 0.9690 |

After reviewing the MPI and other indices, it was found that the t2 section showed a relatively large increase, and productivity was secured with MPI above 1 until the t3 point. However, it was found that this was not the case from t4 to t6. Apart from the individual units of efficiency, it is judged that productivity has decreased over time. Looking at the overall flow, productivity would improve if the efficiency for technical improvement increased, rather than solving operational problems.

## 5. Conclusions

This study analysed the importance of each field in the review of local financial investment projects in Korea and prioritised it to analyse the operational performance of priority fields. Local financial investment projects have a significant impact on the local finance and quality of life of local residents. Moreover, these serve as critical factors in the operation of local finance because the amount of accompanying resources is huge due to the nature of projects. In recent years, the importance of insolvency at the stage of business operations, not until the completion of the project, is also emerging.

Therefore, this study sorted out the projects chosen through industry-related tables and existing investment review history management and constructed a hierarchical structure diagram based on previous domestic and foreign studies. Through this, a total of 14 major fields were derived, and FGI was performed on experts to finalise and prioritise projects. Efficiency and productivity were measured in 17 cities and provinces in Korea, and each field's efficiency and overall productivity analysis over time were presented, along with a short-term analysis. As a result of the analysis, priority areas were determined by investment screening in the order of social welfare (0.110), healthcare (0.108), fire prevention (0.097), environmental hygiene (0.092), and culture and tourism (0.088). This indicates

a change in immediate need between before and after COVID-19. Social welfare and healthcare fields were selected as the most important fields considering balance and equity as the top priority. This result indicates the need for welfare and medical facilities for the vulnerable class, and thus the importance of those fields is expected to increase further in the future. Additionally, fields such as fire prevention and environmental hygiene can be regarded as more important than before given the growing attention to infectious diseases or safety issues. We analysed the efficiency (90%) of most of the areas that were recognised as important while in 2020, the efficiency of major areas was reduced to less than 90% due to the income-to-expenditure imbalance from COVID-19 disaster support funds and vaccine purchases. It is believed that resources can be utilised efficiently if the efficiency level similar to this analysis results can be secured by utilising only about 90% of the current input. Finally, when checking the rate of change in the Malmquist productivity index, it was found that the most reasonable way to maximise productivity is to increase efficiency by improving technology improvement rather than operation.

This study provides basic data to ensure sustainability, stable resource distribution, and efficient use of resources by deriving major areas through the local financial investment review and examining the operational performance of the field. This study has limitations because of the inconsistent purpose and direction of each field, the difference between economic and financial performance, and the application of the same time point for the same variables. In the future, follow-up studies considering various aspects will be conducted along with revising and supplementing the limitations mentioned above.

**Author Contributions:** Methodology, H.S.; Validation, H.S.; Formal analysis, J.K.; Data curation, H.S.; Writing—original draft, H.S. and J.K.; Writing—review & editing, J.K.; Supervision, J.K. All authors have read and agreed to the published version of the manuscript.

**Funding:** This work was supported by a research grant from the Kongju National University in 2022.

**Institutional Review Board Statement:** Not applicable.

**Informed Consent Statement:** Not applicable.

**Data Availability Statement:** Not applicable.

**Conflicts of Interest:** The authors declare no conflict of interest.

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
