# Peer review of "A Study on Project Prioritisation and Operations Performance Measurements by the Analysis of Local Financial Investment Projects in Korea"

_sustainability, doi:10.3390/su15075972_

Round 1

Reviewer 1 Report

The paper is well rewritten and well organized. I have 3 main comments which I hope can help the authors further improve their work.

11.     Method. The authors should explain how the categorization is done and what criterion they apply, for the hierarchical structure they present in Figure 1. First, it is not very clear how projects are divided into Economics infrastructure and Social Infrastructure. For example, why is water supply and sewage considered an economics infrastructure while fire prevention a social infrastructure? Second, it is also not very clear how division is done from Class 1 to Class 2. For example, what do the author mean by writing “1.Equity 2. Effectiveness”? Do those numbers (1 and 2) indicate importance levels or not?

22.       Concept. Social Welfare is considered a very important and widely used concept in welfare analysis. It is not very clear whether what the authors refer to as “social welfare” is essentially the same as the standard definition of social welfare in the field of economics and other social sciences. For example, the authors put “Social welfare” into “Social infrastructure” rather than “Economics infrastructure”. The authors may refer to Kuang et al. (2020) for the use of social welfare in the transportation industry, Feng et al. (2020) for retirement policy, Lian et al. (2021) for international trade, Han et al. (2022) for the education industry, and He et al. (2023) for the platform economics, among many others.

33.       Writing. The authors tend to prefer putting a lot of contents into a long paragraph, and this is not reader friendly. For example, line 31-71 can be divided into at least two paragraphs. The same suggestion applies to line 121-161, line 181-264, line 356-392, among possibly others. In addition, there are some expressions that can be improved. For example, line 181-182, “Until now, research that … have scarce.”.

References:

[1]       Zhenhua Feng et al. “Flexible or Mandatory Retirement? Welfare Implications of Retirement Policies for a Population with Heterogeneous Health Conditions”, International Review of Economics and Finance, September 2020, 69: 1032-1055

[2]       Haipeng Han et al. “Online or Face-to-Face? Competition among MOOC and Regular Education Providers,” International Review of Economics and Finance, July 2022, 80: 857-881

[3]     Wei He et al., “Switching cost, Network Externality and Platform Competition,” International Review of Economics and Finance, March 2023, 84: 428-443

[4]       Zhonghong Kuang et al. “Serial and Parallel Duopoly Competition in Multi-Segment Transportation Routes”, Transportation Research Part E: Logistics and Transportation Review, January 2020, 133: 101821

[5]       Zeng Lian et al. “International Trade with Social Comparisons”, Review of International Economics, August 2021, 29(3): 533-556

Author Response

Response to Reviewer 1 Comments

Point 1: Method. The authors should explain how the categorization is done and what criterion they apply, for the hierarchical structure they present in Figure 1. First, it is not very clear how projects

are divided into Economics infrastructure and Social Infrastructure. For example, why is water supply and sewage considered an economics infrastructure while fire prevention a social infrastructure? Second, it is also not very clear how division is done from Class 1 to Class 2. For example, what do the author mean by writing “1.Equity 2. Effectiveness”? Do those numbers (1 and 2) indicate importance levels or not?

Response 1: The contents of the hierarchical structure diagram have been revised and supplemented in each step. The numbers 1 and 2 are additionally stated in the text as indicating the level of importance.

Point 2: Concept. Social Welfare is considered a very important and widely used concept in welfare analysis. It is not very clear whether what the authors refer to as “social welfare” is essentially the same as the standard definition of social welfare in the field of economics and other social sciences. For example, the authors put “Social welfare” into “Social infrastructure” rather than “Economics infrastructure”. The authors may refer to Kuang et al. (2020) for the use of social welfare in the transportation industry, Feng et al. (2020) for retirement policy, Lian et al. (2021) for international trade, Han et al. (2022) for the education industry, and He et al. (2023) for the platform economics, among many others.

Response 2: The social welfare field was added based on the criteria of the Korea Industrial Association Table. This is a table that distinguishes the most widely used business areas in Korea. The description of the hierarchical diagram further describes this content.

Point 3: Writing. The authors tend to prefer putting a lot of contents into a long paragraph, and this is not reader friendly. For example, line 31-71 can be divided into at least two paragraphs. The same

suggestion applies to line 121-161, line 181-264, line 356-392, among possibly others. In addition, there are some expressions that can be improved. For example, line 181-182, “Until now, research that … have scarce.”

Response 3: As advised, the paragraph has been rescheduled in detail. Thank you very much.

Reviewer 2 Report

The study aims to prioritise investment projects of local Korean governments through to secure sustainability by measuring the operational performances of these projects recurring to AHP technique to draw relative weight, followed by their correction, thereby deriving priorities across all fields. Since important areas prioritise balance and equity and their operational efficiency and productivity, for the measurement, the authors used six years of time series data from 2015 to 2020, and single point-in-time and annual trend analysis to conduct DEA analyses. The input factors were the amount of revenue and the number of public officials, and the output factors included the water and sewage supply rate, the number of tourists traveling for free, and the number of hospitals, welfare facilities, and rescue cases. Results revealed that the operating performance was excellent. However, to secure additional operational efficiency and productivity, technological innovation is needed.

1) General comment:

the paper discusses the priorities of investment projects of local Korean governments deriving policy implications that are specific to te Korean framework, without particularly innovative methodologies or applications of the same. I suggest to better motivate the analysis, his relevance for international readers (in line with the international diffusion of the journal) and the contribute to the existing scientific literature. The main novelty of the paper is to combine DEA and AHP, however DEA is an econometric tool to estimate production functions, something "neutral" to experts’ opinion, AHP results are subjective. Indeed, Once the hierarchy is built, the decision makers systematically evaluate its various elements by comparing them to each other two at a time, with respect to their impact on an element above them in the hierarchy. In making the comparisons, the decision makers can use concrete data about the elements, but they typically use their judgments about the elements' relative meaning and importance. What is the advantage to combine objective results (DEA) with expert personal judgment in this study?

2) Regarding the methodology adopted I will separate my comments on AHP and DEA analyses.

2a) AHP analysis:

Remembering that AHP, rather than prescribing a "correct" decision, helps decision makers find one that best suits their goal and their understanding of the problem, it could be useful to better illustrate the goal of the expert involved. Otherwise, the research will appear biased by their opinions.

In this sense, I would spend more space in better describing the pool of experts involved. It includes 42 university professors (41.2%), 33 from government-funded research institutes (32.4%), 15 from private architectural firms or engineering companies (14.7%), and 12 other expert groups or public officials  (11.7%). Some of them may be closer to the government than others and, since some of the step of AHP are based on their opinions, the paper would be improved by some comments on this issue.

In general I would suggest to completely review section 3, describing in details, step-by-step the various phases of the study.

2b) DEA analysis:

I have some minor comments about DEA. The number of public officials (persons)and revenues are considered as input factors. The author state that this choice follows other studies, without citing them. I would suggest to better motivate this choice. Furthermore, it is demonstrated that, in the presence of non-constant returns to scale, the Malmquist productivity index (MPI) does not accurately measure productivity change. The bias is systematic and depends on the magnitude of scale economies. See E. Grifell-Tatjé, C.A.K. Lovell, A note on the Malmquist productivity index, Economics Letters, Volume 47, Issue 2, 1995. It is not clear if the estimated MPI in table 9 is based on CRS or VRS. In addition, please explain the meaning of DRS and IRS in table 8.

3) Please revise the conclusions according to changes that will be made following in the previous points.

Author Response

Response to Reviewer 2 Comments

Point 1: the paper discusses the priorities of investment projects of local Korean governments deriving policy implications that are specific to te Korean framework, without particularly innovative

methodologies or applications of the same. I suggest to better motivate the analysis, his relevance for international readers (in line with the international diffusion of the journal) and the contribute to the existing scientific literature. The main novelty of the paper is to combine DEA and AHP, however DEA is an econometric tool to estimate production functions, something "neutral" to experts’ opinion, AHP results are subjective. Indeed, Once the hierarchy is built, the decision makers systematically

evaluate its various elements by comparing them to each other two at a time, with respect to their impact on an element above them in the hierarchy. In making the comparisons, the decision makers can use concrete data about the elements, but they typically use their judgments about the elements' relative meaning and importance. What is the advantage to combine objective results (DEA) with expert personal judgment in this study?

Response 1: We tried to determine whether the personal judgment of experts is actually being derived as an objective result. It is true that the two methodologies are different from each other, but by combining them, we tried to compare the current investment business review in Korea and the actual operation. The current investment business focuses on sustainability. Therefore, I would appreciate it if you could think that the qualitative part was considered together with the quantitative part that can support it. Lastly, we separated AHP and DEA according to your opinion.

Point 2: Remembering that AHP, rather than prescribing a "correct" decision, helps decision makers find one that best suits their goal and their understanding of the problem, it could be useful tobetter illustrate the goal of the expert involved. Otherwise, the research will appear biased by their opinions.

In this sense, I would spend more space in better describing the pool of experts involved. It includes 42 university professors (41.2%), 33 from government-funded research institutes (32.4%), 15 from private architectural firms or engineering companies (14.7%), and 12 other expert groups or public

officials (11.7%). Some of them may be closer to the government than others and, since some of the step of AHP are based on their opinions, the paper would be improved by some comments on this issue.In general I would suggest to completely review section 3, describing in details, step-by-step the various phases of the study.

Response 2: Based on your opinion, I reflected the explanation in addition to the expert group. In addition, the analysis results were subdivided and presented step by step. In particular, the hierarchy diagram details the basis for setting each layer. Thank you very much for your detailed advice.

Point 3: I have some minor comments about DEA. The number of public officials (persons)and revenues are considered as input factors. The author state that this choice follows other studies, without citing them. I would suggest to better motivate this choice. Furthermore, it is demonstrated that, in the presence of non- constant returns to scale, the Malmquist productivity index (MPI)

does not accurately measure productivity change. The bias is systematic and depends on the magnitude of scale economies. See E. Grifell-Tatjé, C.A.K. Lovell, A note on the Malmquist productivity index, Economics Letters, Volume 47, Issue 2, 1995. It is not clear if the estimated MPI in table 9 is based on CRS or VRS. In addition, please explain the meaning of DRS and IRS in table 8

Response 3: Based on the advice, 5 studies were cited and written in the text. MPI is a method of estimating the growth rate of total factor productivity. It is defined as the index of the output for the input element based on the distance function without assuming a specific production function. Although it is not accurate in this respect, I would appreciate it if you could understand it as an analysis to find out the operation status. Finally, it was written that it is an MPI based on CRS.The meaning of VRS, DRS, and IRS is additionally stated in the table. Thank you.

Round 2

Reviewer 2 Report

Authors made changes to the paper following my suggestions.